# Nonreciprocal charge transport up to room temperature in bulk Rashba semiconductor α-GeTe

Yan Li[1,6], Yang Li[2,3,6], Peng Li [1], Bin Fang[1], Xu Yang[2,3], Yan Wen[1], Dong-xing Zheng[1], Chen-hui Zhang [1], Xin He[1], Aurélien Manchon [1,4], Zhao-Hua Cheng [2,3,5 ✉] & Xi-xiang Zhang [1 ✉]

Nonmagnetic Rashba systems with broken inversion symmetry are expected to exhibit nonreciprocal charge transport, a new paradigm of unidirectional magnetoresistance in the absence of ferromagnetic layer. So far, most work on nonreciprocal transport has been solely limited to cryogenic temperatures, which is a major obstacle for exploiting the room-temperature two-terminal devices based on such a nonreciprocal response. Here, we report a nonreciprocal charge transport behavior up to room temperature in semiconductor α-GeTe with coexisting the surface and bulk Rashba states. The combination of the band structure measurements and theoretical calculations strongly suggest that the nonreciprocal response is ascribed to the giant bulk Rashba spin splitting rather than the surface Rashba states. Remarkably, we find that the magnitude of the nonreciprocal response shows an unexpected non-monotonical dependence on temperature. The extended theoretical model based on the second-order spin–orbit coupled magnetotransport enables us to establish the correlation between the nonlinear magnetoresistance and the spin textures in the Rashba system. Our findings offer significant fundamental insight into the physics underlying the nonreciprocity and may pave a route for future rectification devices.

[1] Physical Science and Engineering Division, King Abdullah University of Science and Technology (KAUST), Thuwal 23955–6900, Saudi Arabia. [2] State Key Laboratory of Magnetism and Beijing National Laboratory for Condensed Matter Physics, Institute of Physics, Chinese Academy of Sciences, Beijing 100190, China. [3] School of Physical Sciences, University of Chinese Academy of Sciences, Beijing 100049, China. [4] Aix-Marseille Univ, CNRS, CINaM, Marseille, France. [5] Songshan Lake Materials Laboratory, Dongguan, Guangdong 523808, China. [6] These authors contributed equally: Yan Li, Yang Li. ✉email: zhcheng@iphy.ac.cn; xixiang.zhang@kaust.edu.sa

The nonreciprocal transport of propagating particles or quasiparticles in noncentrosymmetric materials has opened up various avenues for research on symmetry-related physical phenomena as well as potential applications in optical isolators, circulators, and microwave diodes over a broad range of frequencies[1–3]. On the basis of symmetry arguments, a striking electrical manifestation of inversion symmetry breaking is the emergence of nonreciprocal charge transport, i.e., inequivalent rightward and leftward currents[4]. Under further breaking time inversion symmetry via applying a magnetic field **B**, non-reciprocal charge transport characterized by the current-direction **I**-dependent nonlinear resistivity $R(\mathbf{B}, \mathbf{I})$ can be expressed as[5–7]

$$R(\mathbf{B}, \mathbf{I}) = R_0\left(1 + \beta \mathbf{B}^2 + \gamma \mathbf{B} \cdot \mathbf{I}\right) \quad (1)$$

where $R_0$, $\beta$, and $\gamma$ are the resistance at zero magnetic field, the coefficient of the normal magnetoresistance, and the non-reciprocal coefficient, respectively. In this context, the non-reciprocal response scales linearly with both the applied electric current and the magnetic field, which has been recently discovered in polar semiconductors[6], topological insulators (TIs)[8], and several interface/surface Rashba systems[9] with spin-momentum locked bands. Unlike magnetoresistance in ferromagnet/heavy metals (FM/HMs) or FM/TI bilayers, in which the FM layer plays an essential role as a source of spin-dependent scattering, the nonreciprocal charge transport in noncentrosymmetric materials without FM layers introduces a new paradigm of unidirectional magnetoresistance (UMR) as a consequence of the second-order response to the electric field[10–14]. Such a UMR sparks a surge of interest in realizing two-terminal rectification, memory, and logic devices[7,14,15]. To date, more efforts to hunt for the materials with larger $\gamma$ values by taking the spin–orbit interaction and Fermi energy into account are being made in interface/surface Rashba systems[9,15,16]. However, given the low Rashba spin splitting energy, e.g., 3 meV ($\sim$35 $k_B$) in LaAlO$_3$/SrTiO$_3$[9], 5 meV ($\sim$58 $k_B$) in Ge(111)[15], nonreciprocal transport can only be observed at very low temperature, and here $\gamma$-value decreases dramatically with increasing temperature due to the thermal fluctuation. To exploit the numerous possible applications in Rashba systems, the preservation of nonreciprocal charge transport at room temperature is vigorously pursued.

Although Rashba effect is commonly associated with low-dimensional systems and heterostructures, the recent discovery of sizeable Rashba splitting in bulk materials has attracted much attention[17–19]. α-GeTe, one of the emergent ferroelectric Rashba semiconductors, has a noncentrosymmetric crystal structure up to a high critical temperature $T_c \sim 700$ K[20–22]. As theoretically predicted and experimentally verified, α-GeTe forms a giant bulk Rashba-type spin splitting with the largest observed Rashba constant up to $\alpha \sim 5$ eV Å and hosts electric field-controlled Rashba-type spin textures as well[23–25]. Furthermore, the corresponding spin splitting energy in α-GeTe, proportional to $\alpha^2$, reaches up to $\sim$200 meV ($\sim$2300 $k_B$)[23], one order of magnitude stronger than the thermal energy $k_B T$ at room temperature, which thus makes it as a prominent platform to realize the nonreciprocal charge transport even at room temperature. In this work, we demonstrate the existence of nonreciprocal charge transport up to 300 K originating from the bulk Rashba states in α-GeTe. Nonreciprocal coefficient $\gamma$ exhibits a non-monotonic dependence with increasing temperature $T$. The physical mechanism underlying the characteristics can be understood by combining the angle-resolved photoelectron spectroscopy (ARPES) measurements and theoretical calculations.

## Results and discussion

**Basic characterizations**. We fabricated the Te-terminated α-GeTe films on Al$_2$O$_3$ (0001) substrates by molecular beam epitaxy (MBE). The rhombohedral crystal structure of α-GeTe (space group R3m) with the displaced adjacent Ge and Te layers is schematically presented in Fig. 1a. Such a noncentrosymmetric structure manifests itself as the ferroelectric order **P** along c axis and the Rashba-type spin–orbit splitting bands[26]. To verify the Rashba-type splitting, we performed in situ ARPES measurements using the photon energy of hν = 21.2 eV. Figure 1b shows the map of electronic band structure along the high-symmetry direction $\overline{\Gamma} - \overline{K}$, in which the Rashba splitting is most pronounced. The energy dispersion of bulk Rashba states presents a momentum splitting along $\overline{\Gamma} - \overline{K}$ with $\Delta k \approx 0.1$ Å$^{-1}$ and the giant Rashba parameter around 4.3 eV Å, in good agreement with other reports and calculations for Te-terminated α-GeTe[23–27]. We also resolve the band-crossing point (BCP) of bulk Rashba state at $\overline{\Gamma}$, which is located $\sim$0.14 eV below the Fermi level. Meanwhile, the surface states crossing the Fermi level at significantly higher momenta are also clearly discernable, in particular their linear dispersions in the vicinity of the $\overline{\Gamma}$ point. In addition, the Fermi level position $\mu$ is nearly independent of the temperature $T$, as shown in Supplementary Fig. 4.

α-GeTe thin film is patterned into Hall devices for transport measurements (see "Methods"). Figure 1c shows the temperature dependence of the resistivity of α-GeTe. Unlike a usual semiconductor, α-GeTe shows a low resistivity $\rho \sim 0.12$ mΩ cm at 300 K and a metallic behavior down to 3 K is observed clearly in its temperature dependence. Hall measurements display a linear dependence of the Hall resistances $R_{xy}$ on the applied magnetic field, a typical ordinary Hall effect in a usual semiconductor with the conventional single carrier, as shown in the inset of Fig. 1d. The positive slope indicates that the dominant carriers are holes (P-type) in α-GeTe films. This is also confirmed via ARPES measurement, which demonstrates that Fermi surface lies in the valence band. As shown in Fig. 1d, the extracted carrier concentration $n$ decreases from $3.0 \times 10^{20}$ cm$^{-3}$ at 300 K to a minimum $2.8 \times 10^{20}$ cm$^{-3}$ at 100 K, then it increases upon further lowering the temperature. Below $\sim$15 K, the carrier concentration reaches a saturation value $2.9 \times 10^{20}$ cm$^{-3}$. Although the carrier concentration shows anomalous temperature dependence from 3 to 300 K, its absolute change in the magnitude remains weak, revealing a negligible shift of Fermi level with temperature. According to previous reports, the metallic resistivity and high P-type carrier concentrations are ascribed to the natural tendency for Ge deficiency in α-GeTe films[28–31]. However, the anomalous temperature-dependent carrier concentration remains puzzling, which will be discussed later.

**Nonreciprocal transport response**. To explore the existence of nonreciprocal transport response, we performed angular-dependent ac harmonic measurements. The first ($R_\omega$) and the second-harmonic resistance ($R_{2\omega}$) upon injecting sinusoidal ac current $\tilde{I}_\omega$ into devices are simultaneously detected during the rotation of the applied magnetic field in three different geometries, as illustrated in Fig. 2a. The rotation angles in $xy$, $yz$, and $zx$ planes are defined with respect to $+x$, $+z$, and $+z$ axis, respectively. Under a fixed strength of the magnetic field $B = 9$ T and at $T = 10$ K, the angle-dependent $R_\omega$ is found to be a sinusoidal curve with a period of 180° in all measurement geometries (see Supplementary Fig. 8), while $R_{2\omega}$ in the $xy$ and $yz$ planes displays a sinusoidal angular dependence with nearly same amplitude and a period of 360°, as shown in Fig. 2b. In contrast, $R_{2\omega}$ is vanishingly small when rotating the magnetic-field angle in $xz$ plane.

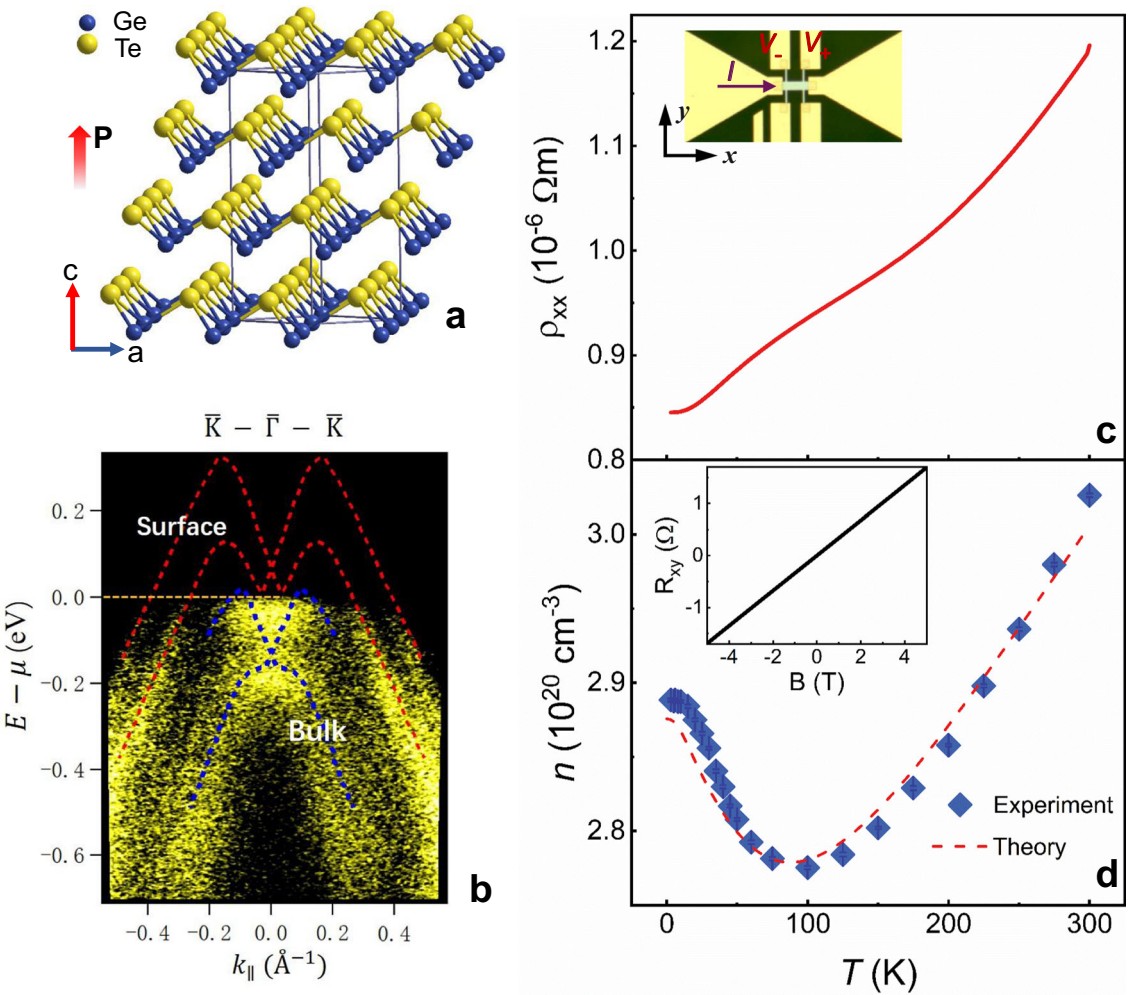

**Fig. 1 Basic characterizations of α-GeTe films. a** Schematic illustration of rhombohedral crystal structure of α-GeTe. **b** ARPES band map along $\overline{K} - \overline{\Gamma} - \overline{K}$ direction measured at 5 K. The red and blue dashed curves trace the surface and bulk Rashba states, respectively. **c** Temperature dependence of the resistivity $\rho_{xx}$. The inset is an optical image of α-GeTe device. **d** Experimental value of the carrier concentration $n$, as well as the theoretical predication, is plotted as functions of temperature. The inset shows a representative example of the Hall resistivity $R_{xy}$ with the magnetic field at 10 K.

These results indicate that $R_{2\omega}$ follows $\mathbf{I} \cdot (\mathbf{P} \times \mathbf{B})$ with the polarization $\mathbf{P}$ along $+z$ direction and reaches its maximum value when $B$ is applied along the $+y$ axis.

As shown in Fig. 2c, even at 300 K, we still observed an angular-dependent $R_{2\omega}$, following $R_{2\omega} = \Delta R_{2\omega}\sin\varphi$ at a fixed injected current density $j = 7.5 \times 10^5$ Acm$^{-2}$ in the $xy$ rotation plane with different magnetic-field strengths. Here, $\Delta R_{2\omega}$ denotes the amplitudes of the angular-dependent $R_{2\omega}$ and $\varphi$ represents the angle between the applied magnetic field and $+x$. Remarkably, $R_{2\omega}$ changes sign when reversing the magnetic field, demonstrating a characteristic UMR. The values of $\Delta R_{2\omega}$ were extracted at varied magnetic field $B$, as plotted in Fig. 2d. The $\Delta R_{2\omega}$ clearly scales linearly with $B$ with a negligible intercept. Moreover, the angular-dependent $R_{2\omega}$ is also measured by varying the current density $j$ at a fixed magnetic field $B = 9$ T. The values of $\Delta R_{2\omega}$ show a linear dependence of $j$, as shown in Fig. 2e. The UMR with a bilinear magnetoresistance characteristic, $\Delta R_{2\omega}$ proportional to both the applied magnetic field $B$ and the injected current $j$, was observed, which unambiguously verifies the existence of the nonreciprocal charge transport in α-GeTe. Similar UMR characteristics were also observed in other nonmagnetic non-centrosymmetric systems with spin-momentum locking, such as BiTeBr[6], WTe$_2$[32], and SrTiO$_3$[9], which are distinct from those in FM/HM and FM/TI systems. The UMR in FM/HM or FM/TI

originates from the spin accumulation at the interfaces induced by the spin Hall effect or the Rashba–Edelstein effect and complies with the chiral rule $(\mathbf{j} \times \hat{\mathbf{z}}) \cdot \mathbf{M}$ with the magnetization $\mathbf{M}$[10–12,33]. Thus, the values of $\Delta R_{2\omega}$ in these heterostructures are also linear with the magnitude of the applied ac current and reach a saturated value at the saturation of $\mathbf{M}$[10,11]. According to the Rashba Hamiltonian $\alpha(\mathbf{k} \times \sigma) \cdot \hat{\mathbf{z}}$, the shift of wave vector $\Delta\mathbf{k}$ upon injecting charge current $\mathbf{j}$ gives rise to the term $\alpha(\Delta\mathbf{k} \times \sigma) \cdot \hat{\mathbf{z}} \sim (\hat{\mathbf{z}} \times \mathbf{j}) \cdot \sigma$, which means that a pseudomagnetic field $\sim\hat{\mathbf{z}} \times \mathbf{j}$ acts on the spins $\sigma$ resulting in the asymmetric spin-dependent scattering (i.e., UMR)[15,34].

Furthermore, the nonreciprocal coefficient $\gamma$ over a temperature range of 3–300 K in α-GeTe can be extracted by $\gamma = 2\Delta R_{2\omega}/(R_0BI)$[6]. Figure 2f reports $\gamma$ as a function of temperature $T$. We note that $\gamma$ has the same order of magnitude over the entire temperature range. Upon increasing temperature, $\gamma$ decreases slightly from $1.60 \times 10^{-3}$ A$^{-1}$T$^{-1}$ at 3 K to a minimum $1.55 \times 10^{-3}$ A$^{-1}$T$^{-1}$ at 50 K, and then increases to $2.34 \times 10^{-3}$ A$^{-1}$T$^{-1}$ at about 200 K. As the temperature increases from 200 to 300 K, $\gamma$ monotonically decreases to $1.95 \times 10^{-3}$ A$^{-1}$T$^{-1}$. The observed temperature-dependent $\gamma(T)$ is remarkably distinct from those reported in other Rashba systems, in which $\gamma$ significantly decreases to a very low value with increasing temperature to only a few tens of Kelvin[6,9,15].

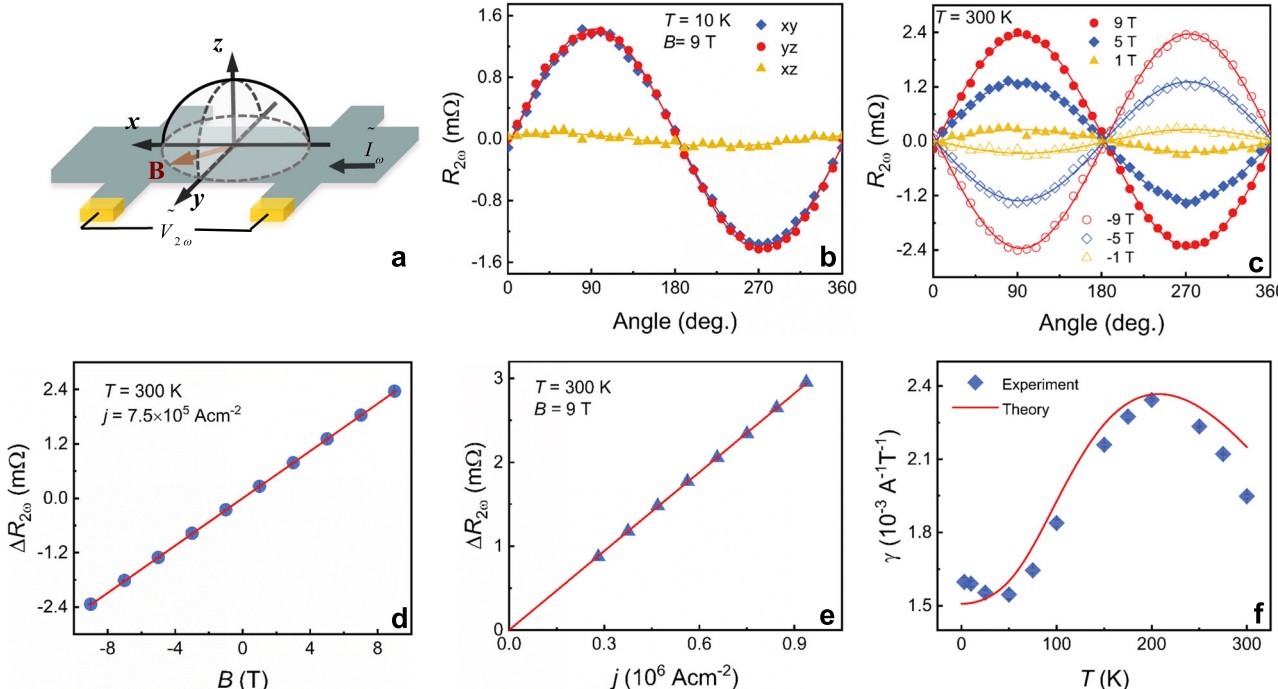

**Fig. 2 Unidirectional magnetoresistance in α-GeTe films. a** Schematic diagram of ac harmonic measurement configurations with ac current applied along x-axis. **b** Dependence of the second-harmonic longitudinal resistance $R_{2\omega}$ on angles in three different geometries with the current density $j = 7.5 \times 10^5 \text{Acm}^{-2}$ at 10 K and 9 T. **c** $R_{2\omega}$ as a function of in-plane magnetic-field angle (xy-scan) at 300 K under various magnetic fields, where the symbols are experimental data and the solid curves are the fits to $\Delta R_{2\omega} \sin\varphi$. **d** Magnetic field-dependent $\Delta R_{2\omega}$ extracted from **c** with $j = 7.5 \times 10^5 \text{Acm}^{-2}$. **e** $\Delta R_{2\omega}$ as a function of the applied ac current amplitude at 300 K and 9 T. **f** Temperature dependence of $\gamma$. The cyan symbols are the experimental results and the red line is from the theoretical predication.

The remarkable trend of $\gamma(T)$ from 3 K to the room temperature was reconfirmed in other α-GeTe films, as shown in Supplementary Fig. 13. This unambiguously indicates that the feature displayed in Fig. 2f is authentic and one of the characteristic properties of α-GeTe. The toy model based on the pseudomagnetic field discussed above is unable to properly reproduce the temperature dependence of $\gamma$[15]. To date, two scenarios have been proposed to explain the temperature-dependent $\gamma(T)$. One is associated with shift of Fermi level driven by temperature, which was used to interpret the temperature-dependent nonlinear magnetotransport observed in semimetal WTe$_2$[32]. The other scenario involves the density of occupied states modulated by the temperature-dependent Fermi–Dirac distribution function[6]. This model could be applied to α-GeTe by considering the fact that the Fermi level position only slightly varies with the temperature, as confirmed via ARPES and carrier concentration measurements. Furthermore, in contrast with most systems investigated previously, α-GeTe possesses both surface and bulk Rashba states; henceforth, their relative contributions to the magnetotransport and in particular to the temperature dependence of $\gamma$ requires further scrutiny. In the following part, the contributions from both the surface and bulk Rashba states to the nonreciprocal charge transport in α-GeTe are theoretically analyzed.

**Theoretical analysis of nonreciprocal charge transport.** On the basis of the fundamental symmetry principles, Onsager reciprocal theorem allows the nonreciprocal response existing in the systems without the inversion symmetry when the time reversal symmetry is also broken[4,5]. Furthermore, the broken inversion symmetry manifests itself as the Rashba-type spin splitting of energy bands with spin-momentum locking[35,36]. Therefore, nonreciprocal charge transport can be investigated using Boltzmann transport

equation in Rashba-type bands[6]. Here, we firstly construct the physical picture for the nonreciprocal transport of α-GeTe by extending the nonlinear second-order spin–orbit coupled magnetotransport model[6,8,37,38]. This model was successfully employed to interpret the bilinear magnetoresistance of TI Bi$_2$Se$_3$ by taking into account of the first- and second-order correction of the carrier distribution. For the sake of simplicity and without loss of generality, we only discuss a 2D Rashba system and neglect the variation of the Rashba constant $\alpha$ with temperature[6]. Figure 3a shows a schematic of the valence band structures with Rashba-type spin splitting in momentum space ($k_x$, $k_y$), described by the Rashba Hamiltonian $h = \frac{\hbar(k_x^2 + k_y^2)}{2m^*} + \alpha(k_x\sigma_y - k_y\sigma_x)$ with $\alpha = 4.3$ eVÅ, the Pauli matrices $\sigma_x$ ($\sigma_y$), and the effective mass of carrier $m^*$. The Fermi surfaces consist of two identical spin helicities above the BCP at zero magnetic field is defined as the zero-point energy), whereas there are two opposite spin helicities below BCP, as illustrated in Fig. 3b, c. When applying an electric field $E_x$, the Fermi contours shift along the $k_x$ direction. The carrier distribution function is expanded as $f = f_0 + f_1 + f_2 + O(k^3)$. Here, $f_0$ is the equilibrium Fermi–Dirac distribution function and $f_n = \left(\frac{e\tau E_x}{\hbar}\frac{\partial}{\partial k_x}\right)^n f_0$ is the $n$-order correction to $f_0$ with the scattering time $\tau$, the elementary charge $e$ and the Planck's constant $\hbar$. The first-order distribution $f_1$ gives rise to the first-order charge current $J_x^1 = e\int\frac{d^2\mathbf{k}}{(2\pi)^2}v(\mathbf{k})f_1(\mathbf{k})$ with group velocity $v(\mathbf{k})$ and Fermi contours subsequently generate an extra imbalance of spin population known as the Rashba–Edelstein Effect[34,39]. In addition, a nonlinear spin current $J_S^2(E_x^2)$ is also simultaneously triggered by the second-order distribution $f_2$ combined with the spin-momentum locking in Rashba bands[40]. Note that the nonlinear spin current $J_S^2(E_x^2)$ due to carriers with energy below the BCP (Fig. 3c) is partially

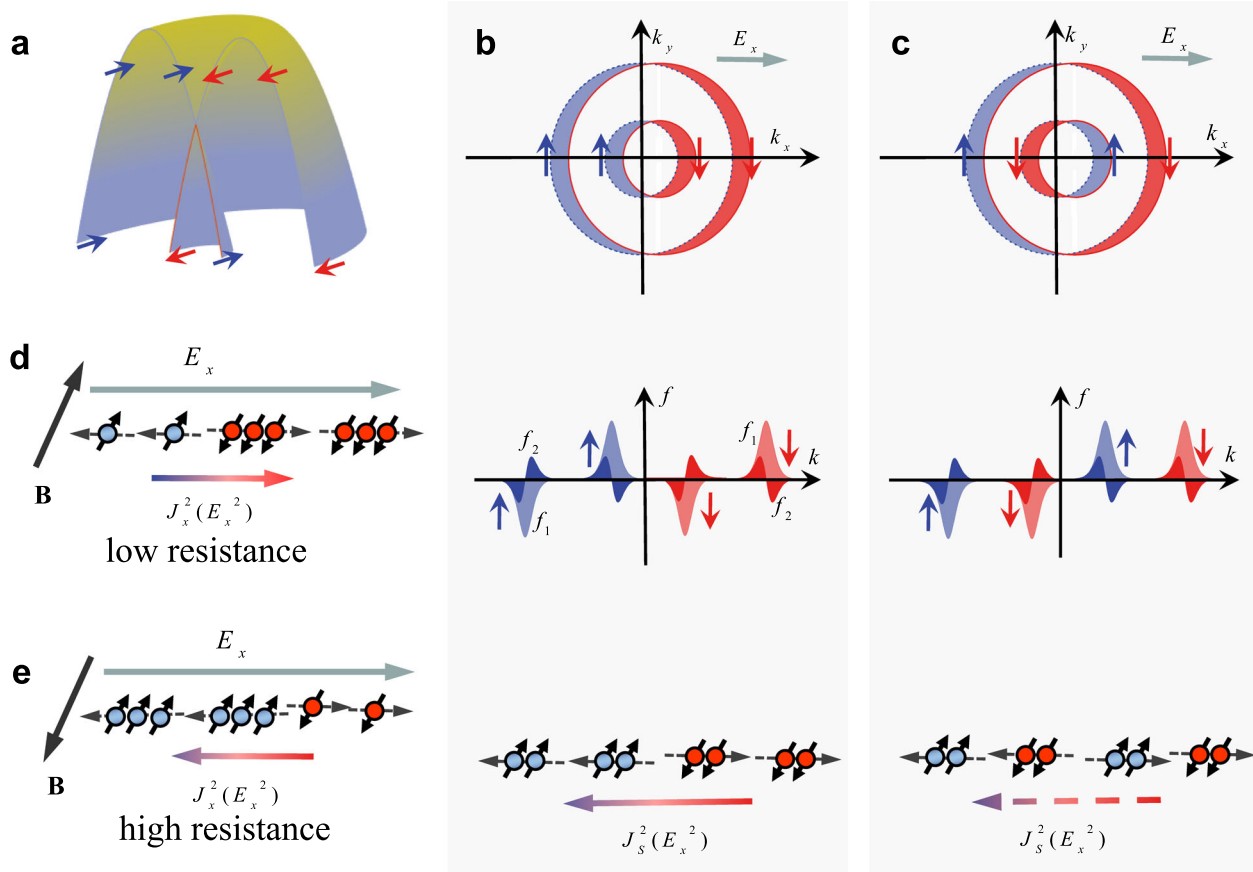

**Fig. 3 The physical figure of UMR in 2D nonmagnetic Rashba systems. a** Sketch of 2D Rashba-type band structures at equilibrium. **b** Top: schematics of Fermi contours (dash circles) above BCP. The outer and inner contours hold the identical spin helicities. The red and purple arrows represent the spin-up and spin-down states, respectively. An applied electric field $E_x$ along $k_x$ axis induces a shift of the Fermi contours (solid circles) toward $k_x$ axis, resulting in an imbalance of carrier occupation on two sides of the Fermi contours. Middle: illustrations of first-order $f_1$ (filled in light red or light purple color codes) and second-order $f_2$ (filled in dark red or dark purple color codes) corrections to the equilibrium distribution function $f_0$ in $E_x$. The parts of $f_1$ and $f_2$ above $k$ axis depict excess of carrier along the arrow direction, whereas other parts below $k$ axis denote depletion of the same. Bottom: the second-order spin current $J_S^2(E_x^2)$ induced by $E_x$. **c** Top: schematics of Fermi contours with $E_x$ (solid circles) and without $E_x$ (dash circles) below BCP. The outer and inner contours depict the opposite spin helicities. The corresponding $f_1$ and $f_2$ is illustrated in middle panel. In this case, $J_S^2(E_x^2)$ is partially canceled due to the opposite spin helicities (bottom panel). **d**, **e** $J_S^2(E_x^2)$ is partially converted into a nonlinear charge current $J_x^2(E_x^2)$ under applied magnetic field **B**. The state with **B** parallel (antiparallel) to the spin polarization direction corresponds to a low (high) resistance state.

compensated because the spin chirality of the inner Fermi contour is opposite to that of the outer one. In contrast, for carriers with energy above the BCP (Fig. 3b), $J_S^2(E_x^2)$ is enhanced due to the cooperative contributions of the identical spin helicities. When an external magnetic field is applied along the $+y$ or $-y$ direction, $J_S^2(E_x^2)$ is partially converted into a magnetic field-direction dependent second-order charge current $J_x^2(E_x^2)$ (denoted by $J_x^2$ below), giving rise to a UMR (i.e., nonreciprocal charge transport), as shown in Fig. 3d–e.

The second-order charge current is further quantified in the following calculations. The Rashba Hamiltonian with in-plane magnetic field $B_y$ oriented along $y$ axis reads as

$$H = \frac{\hbar(k_x^2 + k_y^2)}{2m^*} + \alpha(k_x\sigma_y - k_y\sigma_x) - \frac{1}{2}g\mu_B B_y \sigma_y \quad (2)$$

Here, $g$ and $\mu_B$ are the Landé $g$-factor and the Bohr magneton, respectively[6,41]. For a given applied magnetic field, the typical band dispersion of the Rashba gas is depicted in Fig. 4a and can be parsed into three energy regions. The applied magnetic field gives rise to an energy shift with a value of $g\mu_B|B_y|$, corresponding

to energy region I ($-\frac{g\mu_B|B_y|}{2} - \frac{m^*\alpha^2}{2\hbar^2} < E < \frac{g\mu_B|B_y|}{2} - \frac{m^*\alpha^2}{2\hbar^2}$). Due to the giant pseudomagnetic field of the Rashba spin–orbit coupling up to ~120 T, the region I is narrow under our experimental magnetic field ~9 T. Furthermore, the inner and outer Rashba Fermi contours host opposite spin helicities below BCP (region III $E < \frac{\hbar^2}{2m^*\alpha^2}(\frac{g\mu_B B_y}{2})^2$), whereas they display identical spin helicities above BCP (region I and region II $\frac{\hbar^2}{2m^*\alpha^2}(\frac{g\mu_B B_y}{2})^2 < E < -\frac{g\mu_B|B_y|}{2} - \frac{m^*\alpha^2}{2\hbar^2}$). The second-order charge current $J_x^2$ is given by the following integral,

$$J_x^2 = e\int \frac{d^2\mathbf{k}}{(2\pi)^2} v(\mathbf{k}) f_2(\mathbf{k}) \quad (3)$$

At a fixed $E_x$, $J_x^2$ is proportional to the experimental $\Delta R_{2\omega}$ ($J_x^2 \propto \Delta R_{2\omega}$) (see Supplementary Information)[32].

Nonreciprocal charge transport manifests itself as a bilinear magnetoresistance, that is, $\Delta R_{2\omega}$ scales linearly with both the injected charge current and the applied magnetic field. The former is implicit in Eq. (3), whereas the latter is numerically calculated in the following. Figure 4b summarizes the dependence

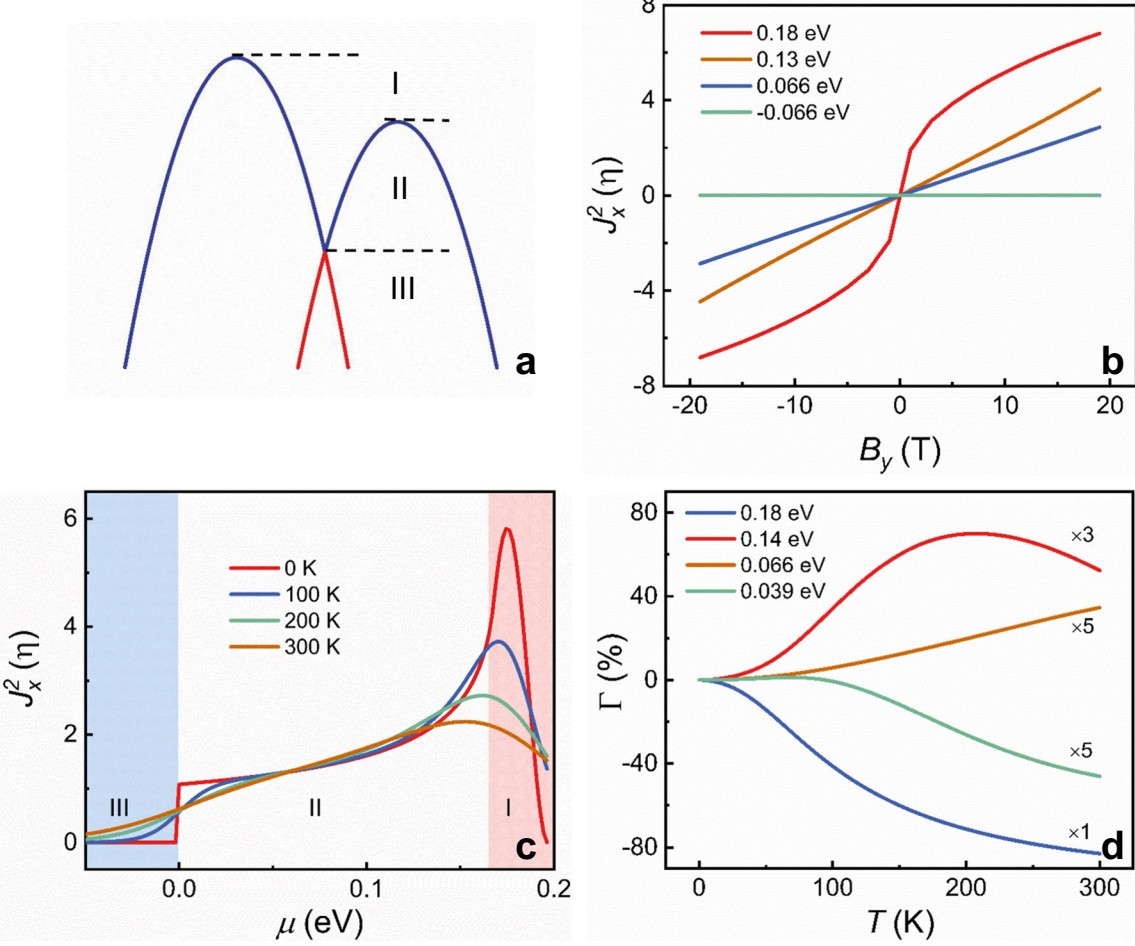

**Fig. 4 Theoretical calculations for second-order charge current $J_x^2$. a** Scheme of 2D Rashba-type bands under the in-plane magnetic field $B_y$. **b** The calculated $J_x^2$ (in unit of $\eta = 0.5m^{*-1}\alpha^{-1}\hbar^{-2}E_x^{-2}e^3\tau^2$) as functions of the applied magnetic field $B_y$ at different Fermi level position at 0 K. **c** Evolution of $J_x^2$ as functions of Fermi level position at $B_y = 10$ T at various temperatures. The three-color codes correspond to the three energy regions in **a**. **d** Temperature dependence of $\Gamma$ at various Fermi levels. For comparison, $\Gamma$ is multiplied by different factors.

of $J_x^2$ as a function of $B$ for various Fermi level positions $\mu$. $J_x^2$ increases linearly with $B$ at $\mu = 0.13$ eV and 0.066 eV (region II). Besides, $J_x^2$ nonlinearly changes with $B$ at $\mu = 0.18$ eV (region I) and $T = 0$ K, and then $J_x^2(B)$ returns to the linear relationship with increasing temperature (see Supplementary Fig. 2). In those cases, the change in sign of $J_x^2$ occurs when reversing magnetic field $B$. In addition, the magnitude of $J_x^2$ is mostly negligible at all magnetic fields with $\mu = -0.066$ eV (region III). Therefore, as described above, the characteristics of bilinear magnetoresistances reappear in the nonlinear second-order spin–orbit coupled magnetotransport model.

To identify the relative contributions of surface and bulk Rashba states in the nonreciprocal charge transport, we calculated the dependence of $J_x^2$ on the Fermi level position $\mu$ based on Eq. (3), as illustrated in Fig. 4c. $J_x^2$ exhibits a significant enhancement with a peak in region I and gradually decreases as Fermi level position shifts toward the lower energy in region II and then fades away with Fermi surface across BCP into region III. It is shown that $J_x^2$ is quite sensitive to $\mu$. In addition, the peak of $J_x^2$ in region I progressively sinks upon increasing the temperature. Note that $J_x^2$ in region III is nearly negligible in comparison with those in region I and II, demonstrating that the contributions of two opposite spin helicities to $J_x^2$ compensate each other. These theoretical results provide a guideline to distinguish the

contributions of surface and bulk Rashba states in the nonreciprocal charge transport. As shown in Fig. 1b, the band structure of α-GeTe includes bulk as well as surface Rashba states. Based on ARPES measurements, the Fermi level lies below BCP for surface Rashba states, but above BCP for bulk Rashba states. As a consequence, it is reasonable to claim that the bulk Rashba rather than surface states are the dominant contribution to the nonreciprocal charge transport in α-GeTe.

To gain further insight into the temperature-dependent $\gamma(T)$, we calculated the dependence of $\Gamma(T) = [\gamma(T) - \gamma(0K)]/\gamma(0K)$, the fractional difference of $\gamma$ at $T$ and at 0 K, as a function of the temperature for various values of $\mu$. To do so, we only consider the temperature-related broadening of the distribution functions. As plotted in Fig. 4d, $\Gamma$ decreases monotonically with increasing temperature at $\mu = 0.18$ eV and $\mu = 0.039$ eV, which has been well documented in previous reports[6,9,15]. In comparison, a novel temperature dependence $\Gamma(T)$ is observed at $\mu = 0.066$ eV, indicating $\gamma$ increases monotonically as temperature increases, which has never been reported before. Assigning $\mu = 0.14$ eV, $\Gamma(T)$ shows a non-monotonic evolution with temperature, indicating that $\gamma$ increases upon increasing the temperature below ~200 K and then decreases as the temperature rises, which is in excellent qualitative agreement with our experimental observations (Fig. 2f). In particular, the calculated Fermi level position $\mu = 0.14$ eV is also comparable with that of ARPES

measurements, as shown in Fig. 1b. Consequently, it can be concluded that the temperature dependence of $\gamma(T)$ derives from the variation of carrier occupation states with temperature and the sensitivity of $J_x^2$ to the Fermi level position. This conclusion is further supported by the consistency between the experimental and theoretical temperature dependence of the carrier concentration, $n = \int \frac{f_0 d^3 \mathbf{k}}{(2\pi)^3}$, as shown in Fig. 1d.

In summary, we unambiguously demonstrated the existence of nonreciprocal charge transport up to room temperature in Rashba semiconductor α-GeTe, in which both the surface and bulk Rashba states exist. The nonreciprocal charge transport yields a UMR with a bilinear magnetoresistance characteristic. More interestingly, we observed an unconventional temperature-dependent nonreciprocal coefficient $\gamma$, in which $\gamma$ increases with raising temperature below 200 K and monotonically decreases in the range of 200–300 K. To understand the physics underlying these observations, a second-order spin–orbit coupled magneto-transport model considering the distinction of the spin chirality in Rashba bands has been developed. The combination of the ARPES measurements and theoretical calculations strongly suggests that the nonreciprocal response originates from the bulk rather than surface Rashba states, and that the unconventional temperature dependence of $\gamma(T)$ is related to the Fermi level position and the second-order correction of the distribution function. Our work offers valuable insight into the nonreciprocal response and provides pathways towards realizing the room-temperature two-terminal spintronic devices.

## Methods

**Sample preparation**. α-GeTe films were fabricated on insulating $Al_2O_3$ (0001) substrates by MBE with a base pressure of $<2 \times 10^{-9}$ mbar. The epi-ready substrate was annealed at 500 °C for 2 h in vacuum before the epitaxy. The deposition was performed using Ge and Te effusion cells set at $T_{Ge} = 1140$ °C and $T_{Te} = 310$ °C with the substrate temperature at 200 °C. Then, the samples were annealed at the deposition temperature for 30 min to improve the crystalline quality of samples.

**ARPES measurements**. The grown samples are transferred into the in situ ARPES chamber with a base pressure lower than $10^{-10}$ mbar. We used the He discharge lamp (He-I α, h$\nu$ = 21.2 eV) as the photon source and then detected photoelectrons using Scienta DA30 analyzer with an energy resolution of 20 meV and angular resolution of 0.5°. ARPES measurements were performed at various temperatures.

**Transport measurements**. The films were patterned into Hall bar devices with the width of 5 ~30 μm (Hall bar with width of 5 μm is used for the harmonic measurement in the main text) by standard photolithography technique and Ar ion milling, and then Ti(10 nm)/Au(50 nm) electrical contacts were deposited via the electron-beam evaporation. The devices were bonded to the horizontal or vertical rotatable sample holders using Al wires and then installed in the Physical Property Measurement System (PPMS, Quantum Design) to perform the electrical properties measurements. Keithley 6221 current source was used to supply sinusoidal ac current with a frequency of 13 Hz. Meanwhile, the in-phase first (0° phase) and out-of-phase ($\frac{\pi}{2}$ degree phase) second-harmonic voltage signals were probed using two Stanford Research SR830 lock-in amplifiers.

## Data availability

The data that support this study are available from the corresponding author upon reasonable request.

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

## Acknowledgements

We thank Dr. Keita Hamamoto and Dr. Toshiya Ideue for their useful discussions. We thank Dr. Aitian Chen for the technical support on preparing the devices. The work reported was funded by King Abdullah University of Science and Technology (KAUST), Office of Sponsored Research (OSR) under the Award numbers CRF-2015-SENSORS-2708 and CRF-2018-3717-CRG7. This work is also supported by the National Key Research Program of China (grant numbers 2016YFA0300701 and 2017YFB0702702), the National Natural Sciences Foundation of China (Grant numbers 52031015, 1187411, and 51427801), and the Key Research Program of Frontier Sciences, CAS (Grant numbers QYZDJ-SSW-JSC023, KJZD-SW-M01, and ZDYZ2012-2).

## Author contributions

Yan Li, Yang Li, Z.H.C., and X.X.Z. conceived and designed the experiments. Yang Li and X.Y. grew the films and performed the ARPES measurements. Yan Li, B.F., and C.H.Z. fabricated the devices. Yan Li, P.L., Y.W., D.X.Z., C.H.Z., and X.H. carried out the transport measurements. Yan Li performed the theoretical calculations with assistance from A.M. Yan Li and Yang Li wrote the manuscript. All authors discussed the results and contributed to the manuscript preparation.

## Competing interests

The authors declare no competing interests.
