## [Peer Review File · Nature Communications]

REVIEWER COMMENTS

Reviewer #1 (Remarks to the Author):

This paper studies the nonreciprocal transport in α -GeTe both experimentally and theoretically. The new finding is that this material shows nonreciprocal effect even at room temperature, which will be useful for applications. Even more, it shows the nontrivial temperature dependence of the nonreciprocity, which is analyzed nicely by a theoretical calculation based on the Boltzmann equation. I would support the publication of this paper in Nature Communications since it provides a solid result and is scientifically sound. Some minor comments are:

1. Line 101: slop  slope
2. Fig. 3 panels b, c are vertically long, and can be confused with panels d, e.
- 3., Line 137, 189 references are better attached to each material

Reviewer #2 (Remarks to the Author):

Nonreciprocal charge transport is now recognized as a unique functionality in noncentrosymmetric systems and also an emergent property reflecting the underlying physics such as spin-orbit interaction, magnetism, and band geometry/topology. In this manuscript, authors studied the nonreciprocal charge transport in bulk Rashba semiconductor GeTe. They report unusual temperature dependence of the nonreciprocal charge transport and clarified that it survives up to room temperature, which is scientifically interesting and important for the application.

I feel that results are worth publishing in Nature Communications. However, I think the authors should further consider and clarify the following point before accepting the manuscript.

1. I would like to confirm whether they consider the effect of thermoelectric effect. When the sample temperature unprecedentedly changes under the applied current, there is a temperature gradient between the sample and substrate. If the magnetic field is applied along the in-plane direction (perpendicular to the current), Nernst effect, which also produces the DC or second harmonic voltage along the current direction, can be expected. Since thermoelectric effect becomes negligibly small at low temperature I agree that data at low temperature purely indicates the nonreciprocal transport. However, to discuss the magnitude of the second harmonic resistance at high temperature region, I think it is better to clarify whether it can be neglected or not.
2. I want to know how large the domain size of this polar crystal. The sample is small enough and can be regarded as the mono-domain sample?
3. Related to the above question, are there any samples which show the opposite polarity and resultant opposite sign of the nonreciprocal charge transport? If not, what usually determines the polarity direction?
4. In the previous work of bulk Rashba semiconductor BiTeBr (Ref. 6), carrier is electron while it is hole in the present study of GeTe. Does this difference affect the sign of the nonreciprocal charge transport?
5. Theoretical model authors considered in this work seems to be similar (or same) as that discussed in Ref. 6. However, authors successfully explained the unusual non-monotonous temperature dependence of the nonreciprocal charge transport while only monotonous behavior is shown in Ref. 6. What is the difference?
6. In Fig. 4 d, authors should clarify the value of the magnetic field, they used for the calculation.

Reviewer #3 (Remarks to the Author):

The authors reported a nonreciprocal charge transport behavior up to room temperature in semiconductor α -GeTe, whose inversion symmetry is broken due to the ferroelectricity. They found that the magnitude of the nonreciprocal response shows an unexpected non-monotonical dependence on temperature. They used the extended theoretical model based on the second-order spin-to-charge interconversion to establish the correlation between the nonlinear magnetoresistance and the spin textures in the Rashba system, and concluded that the temperature dependence of $\gamma(T)$ derives from the variation of carrier occupation states with temperature and the sensitivity of J_x^2 to the Fermi level position. The investigation is interesting, and the theoretical and experimental results seem reasonable. A few comments and doubts are listed in the following:

- 1) The authors emphasized that the nonreciprocal transport arises from bulk Rashba SOC, not the surface Rashba SOC, which was inferred by the theoretical analysis and the ARPES measurements. Could the authors provide more solid proof to support this opinion, for example, the authors can decorate the surface and then do the measurements, and compare the nonreciprocal transport behavior of the different surfaces.
- 2) In the SI, from line 106-109, "It is seen that the Fermi level position and the Rashba constant in α -GeTe vanishingly vary with the temperature, as shown in Fig. S4, which is consistent with the temperature dependent carrier concentration and the recent report". However, in Fig. 1(d), we can clearly observe the nonlinear change of the carrier concentration vs. temperature, which plays the key role in the non-monotonical behavior.
- 3) Since "the Fermi level position and the Rashba constant in α -GeTe vanishingly vary with the temperature", The authors neglected the temperature dependence of Rashba SOC for simplicity. Although Rashba SOC itself is vanishingly vary with the temperature, is it possible that some physical quantity is sensitive to Rashba SOC. As the authors pointed, the Fermi level is also vanishingly vary with the temperature, however, the sensitivity of J_x^2 to the Fermi level position plays the critical role in the non-monotonical dependence on temperature. If the author could reveal the nonreciprocal transport behavior from 0K to T_c of GeTe (~ 700 K), the investigation would be of more significance and instructive.
- 4) The author should check all the formula in the manuscript and SI, there are many missing symbols, for example, line 44, 68, 89.....
- 5) On line 415, there is a space before the word "induces".

This document contains our responses to the reviewers' comments regarding the manuscript NCOMMS-20-34418, entitled "Nonreciprocal charge transport up to room temperature in bulk Rashba semiconductor α -GeTe". We would like to thank the reviewers very much for the very detailed and constructive comments, which indeed help improve the quality of manuscript. We have addressed all the comments as outlined in detail below.

Reviewer #1 (Remarks to the Author):

This paper studies the nonreciprocal transport in α -GeTe both experimentally and theoretically. The new finding is that this material shows nonreciprocal effect even at room temperature, which will be useful for applications. Even more, it shows the nontrivial temperature dependence of the nonreciprocality, which is analyzed nicely by a theoretical calculation based on the Boltzmann equation. I would support the publication of this paper in Nature Communications since it provides a solid result and is scientifically sound. Some minor comments are:

Response: We greatly appreciate your positive comments and recommendation to publish our manuscript in Nature Communications. It is very encouraging that our work is perceived in such a positive way. We have revised the original manuscript according to your comments.

1. Line 101: slop  slope

Response: Thank you. The word has been corrected in the revised manuscript.

2. Fig.3 panels b,c are vertically long, and can be confused with panels d.e.

Response: Thank you. Following your suggestion, we have revised the panels b and c in Fig. 3. Two light grey boxes have been inserted to make the figure more transparent.

3. Line 137,189 references are better attached to each material.

Response: Thank you. Following your suggestion, we have attached the related references to each material in the revised manuscript.

Reviewer #2 (Remarks to the Author):

Nonreciprocal charge transport is now recognized as a unique functionality in noncentrosymmetric systems and also an emergent property reflecting the underlying physics such as spin-orbit interaction, magnetism, and band geometry/topology. In this manuscript, authors studied the nonreciprocal charge transport in in bulk Rashba semiconductor GeTe. They report unusual temperature dependence of the nonreciprocal charge transport and clarified that it survives up to room temperature, which is scientifically interesting and important for the application.

I feel that results are worth publishing in Nature Communications. However, I think the authors should further consider and clarify the following point before accepting the manuscript.

Response: We appreciate your very encouraging comment on our paper as “They report unusual temperature dependence of the nonreciprocal charge transport and clarified that it survives up to room temperature, which is scientifically interesting and important for the application”. We also thank you for recommending publication after addressing the raised issues. We believe we have well addressed your concerns and improved the manuscript.

1. I would like to confirm whether they consider the effect of thermoelectric effect. When the sample temperature unprecedentedly changes under the applied current, there is a temperature gradient between the sample and substrate. If the magnetic field is applied along the in-plane direction (perpendicular to the current), Nernst effect, which also produce the DC or second harmonic voltage along the current direction, can be expected. Since thermoelectric effect becomes negligibly small at low temperature I agree that data at low temperature purely indicates the nonreciprocal transport. However, to discuss the magnitude of the second harmonic resistance at high temperature region, I think it is better to clarify whether it can be neglected or not.

Response: Thank you for this meaningful comment. As you pointed out, there is a possible thermoelectric effect (Nernst effect) induced by the temperature gradient (∇T) between the sample and substrate upon injecting charge current. The voltage from the Nernst effect is proportional to $\mathbf{B} \times \nabla T$, which could also result in a second-harmonic voltage ($V_{2\omega}$). The contribution from the thermoelectric effect can be identified by measuring the in-plane magnetic field-angle dependent second-order harmonic

transverse ($V_{2\omega}^{xy}$) and longitudinal ($V_{2\omega}^{xx}$) voltages [*Phy. Rev. Lett.* 123, 016801 (2019), *Nat. Commun.* 10, 4510 (2019)]. In our case, as shown in Fig. R1, it is seen that $R_{2\omega}^{xy}$ doesn't nearly manifest $\cos\phi$ or $\sin\phi$ behaviors with $R_{2\omega}^{xy}/R_{2\omega}^{xx} \ll w/l$ (w and l are the length and width of the Hall bar, respectively) at both low and high temperatures, indicating negligible contribution from the thermoelectric effect. In the original manuscript, we only show the measured $V_{2\omega}^{xy}$ at 10 K. Following your concerns, we further added the measured results at various temperatures in Fig. S9 in the revised Supplementary Information.

Fig. R1 Angular dependent second-order harmonic longitudinal (Rxx) and transverse (Rxy) resistances with the current density $j = 7.5 \times 10^5 \text{ Acm}^{-2}$ and magnetic field $B = 9 \text{ T}$ at different temperatures in the α -GeTe with a thickness of 64 nm during rotating the magnetic field in the xy plane.

2. I want to know how large the domain size of this polar crystal. The sample is small enough and can be regarded as the mono-domain sample?

Response: For the epitaxial α -GeTe(0001) film [termed as α -GeTe(111) in Rhombohedral cell choice, α -GeTe(0001) in Hexagonal cell choice, *J. Phys. Chem. C* 116, 15801–15811 (2012)], there is a preferential ferroelectric order with the distortion axis along c -axis. It is clarified that the epitaxial α -GeTe film grown via MBE technique is single domain over several $100 \mu\text{m}^2$, and the polarity can be sustained over the thickness of several hundred nanometers [*Adv. Mater.* 28, 560-565 (2016), *Phys. Rev. B* 94, 201403(R) (2016), *Crystals* 9, 335 (2019)]. This stable domain is present everywhere in the films and only interrupted by stripe like domains with oblique distortion axis, which is confirmed that this structure does not randomize the ferroelectric order [*Crystals* 2019, 9, 335]. Meanwhile, the self-screening process of free carriers provided by intrinsic Ge vacancies in GeTe films cancels the

depolarization field according to first principles calculations and results in a pronounced stability of ferroelectricity in thin films α -GeTe [*Nano Lett.* 11, 1147–1152 (2011), *App. Phys. Lett* 113, 232903 (2018)]. Therefore, the used Hall bar devices with $5\ \mu\text{m} \times 30\ \mu\text{m} \times 64\ \text{nm}$ in the main text can be regarded as single domain.

To further verify our claim, we have also measured the nonreciprocal charge transport in the devices with the size of $0.5\ \mu\text{m} \times 1.5\ \mu\text{m} \times 64\ \text{nm}$, which is fabricated using the e-beam lithography, as shown in Fig. R2. The nonreciprocal charge transport up to room temperature with non-monotonical γ vs. T reappears in the smaller device. We have added the measured results in Fig. S16 in the revised Supplementary Information.

Fig. R2 **a** In-plane angle-dependent second-order harmonic longitudinal resistance ($R_{2\omega}$) at varied magnetic fields at 300 K. The inset is the optical image of the Hall device. **b** Temperature dependent nonreciprocal coefficient γ with the theoretical predication.

3. Related to the above question, are there any samples which show the opposite polarity and resultant opposite sign of the nonreciprocal charge transport? If not, what usually determines the polarity direction?

Response: Thank you for this very valuable comment about the effect of the polarity on the resultant nonreciprocal transport. It's a pity for us that the opposite sign of charge transport due to the opposite polarity is not observed in any measured samples. As shown in Fig. R3, given that the Te-terminated surface is preferred for α -GeTe due to its low surface energy, the strong electronegativity of the surface Te atoms results in a net unipolar polarization at the surface [*Phys. Rev. B* 94, 205111 (2016), *J. Phys. Chem. C* 116, 15801 (2012), *Adv. Mater.* 28, 560-565 (2016)]. However, α -GeTe shows a high tendency to form Ge vacancies that giving rise to p-type charge carrier

concentration. The co-existence of metallicity with the presence of ferroelectricity is very rare in any material system [*Nat. Commun.* 8, 15033, (2017)]. The large concentration of the free charge carriers in α -GeTe screens the applied electric field inhibiting polarization reversal and results in high dielectric loss. This calls into question about the possibility to switch the ferroelectric state in such a “conducting” material. Even so, highly motivated to demonstrate that the spin texture is controllable and switchable via an electric field in α -GeTe, we have also attempted to determine the polarity-direction dependent nonreciprocal charge transport via applying a gate voltage many times. Unfortunately, it is extremely difficult to engineer the polarity due to the high leakage, and the devices are damaged by the high gate voltage. However, the main future objective from a materials design point of view will therefore be to expand the class of ferroelectric Rashba semiconductors beyond α -GeTe in the aim of identifying a “strong” (i.e. not-leaky) ferroelectric, where the full-reversal of the spin-texture via an electric field switching is predicted and offered to experiments for confirmation.

[Redacted]

Fig. R3 Sketch of ferroelectric Te-terminated α -GeTe along c-axis. [*Adv. Mater.* 28, 560-565 (2016), *Nano Lett.* 18, 2751 (2018)]

Fig. R4 Some devices damaged by the high gate voltage or ionic liquid in the measurements.

4. In the previous work of bulk Rashba semiconductor BiTeBr (Ref. 6), carrier is

electron while it is hole in the present study of GeTe. Does this difference affect the sign of the nonreciprocal charge transport?

Response: Thank you for this comment. If the Fermi level position moves to the valence band top from the conduction band bottom, the spin textures of electrons reverse, as shown in Fig. R5 a. The corresponding low and high resistance states also reverse, as shown in Fig. R5 b-c and b'-c'. The types of carriers depend on the downward or upward Rashba-type bands. As a consequence, the types of carriers also affect the sign of the nonreciprocal charge transport.

Fig. R5 a Schematic illustration of the Rashba-type conduction and valence bands including the spin textures. **b** and **c** indicate the low and high resistance states when Fermi level locates at between the band-crossing point and the conduction band bottom, respectively. **b'** and **c'** indicate the high and low resistance states when Fermi level locates at between the valence band top and the band-crossing point, respectively. For comparison, all light red or light purple circles represent electrons.

5. Theoretical model authors considered in this work seems to be similar (or same) as that discussed in Ref. 6. However, authors successfully explained the unusual non-monotonous temperature dependence of the nonreciprocal charge transport while only monotonous behavior is shown in Ref. 6. What is the difference?

Response: Thank you for this comment. In our work, we indeed attempt to explain the unusual temperature dependence of the nonreciprocal charge transport in GeTe using the theoretical model employed in Ref.6. We further extend the theoretical model to a vivid physical figure, as shown in Fig. 3 in the main text. Importantly, we have pointed out that the nonreciprocal coefficient vs. temperature (γ vs. T) is very sensitive to the Fermi level position. As shown in Fig. 4d in the main text, the

evolutions of the nonreciprocal coefficient with temperature show a different behavior at various Fermi level positions. In addition to the monotonous behavior shown in Ref.6, the non-monotonous ones also appear. In reality, in the calculation part of Ref. 6, the non-monotonous behaviors also exist at $\mu < 0$, but are inconspicuous, as shown in Figure 4b and c in Ref.6. Essentially, besides the Rashba constant and the type of carriers, the Fermi level position is quite different between GeTe and BiTeBr, in turn reflected by the temperature dependent carrier concentration. We can confirm the difference of the temperature dependent carrier concentration by comparing the derivative of the temperature dependent resistivity, as shown in Fig. R6.

Fig. R6 a shows temperature dependence of the resistivity ρ_{xx} , the derivative of resistivity $\frac{d\rho_{xx}}{dT}$, and carrier concentration n in GeTe, and the data of **b** is from Ref. 6.

6. In Fig. 4 d, authors should clarify the value of the magnetic field, they used for the calculation.

Response: Thank you for pointing out the missing information in the original manuscript. We have added the value of the magnetic field in Fig. 4c in the revised manuscript, “Evolution of J_x^2 as functions of Fermi level position at $B_y=10$ T at various temperatures.” Besides, Γ (or γ) is independent of the magnetic field according to Eq. (1).

Reviewer #3 (Remarks to the Author):

The authors reported a nonreciprocal charge transport behavior up to room temperature in semiconductor α -GeTe, whose inversion symmetry is broken due to the ferroelectricity. They found that the magnitude of the nonreciprocal response shows an unexpected non-monotonical dependence on temperature. They used the extended

theoretical model based on the second-order spin-to-charge interconversion to establish the correlation between the nonlinear magnetoresistance and the spin textures in the Rashba system, and concluded that the temperature dependence of $\gamma(T)$ derives from the variation of carrier occupation states with temperature and the sensitivity of J_x^2 to the Fermi level position. The investigation is interesting, and the theoretical and experimental results seem reasonable. A few comments and doubts are listed in the following:

Response: Thank you very much for the important suggestions that have helped improve the quality of our paper a great deal. We hope that our corrections fully address your questions. The related issues are elaborated in the following.

1) The authors emphasized that the nonreciprocal transport arises from bulk Rashba SOC, not the surface Rashba SOC, which was inferred by the theoretical analysis and the ARPES measurements. Could the authors provide more solid proof to support this opinion, for example, the authors can decorate the surface and then do the measurements, and compare the nonreciprocal transport behavior of the different surfaces.

Response: Thank you for the insightful suggestion. Following your suggestion, we carried out the harmonic measurements in Hall bar devices (40 nm GeTe films) with different surfaces (bare, 2 nm TiO₂, and 3 nm SiO₂). It is seen that the nonreciprocal charge transport up to room temperature with non-monotonical γ vs. T is independent of the different surfaces, such as air, TiO₂ and SiO₂, as shown in Fig. R7. The results further support our conclusion that the nonreciprocal transport arises from bulk Rashba SOC rather than the surface Rashba states. We have added the results in Fig. S15 in the revised Supplementary Information.

Fig. R7 Evolution of the nonreciprocal coefficient γ with temperature for samples with different surfaces.

2) In the SI, from line 106-109, “It is seen that the Fermi level position and the Rashba constant in α -GeTe vanishingly vary with the temperature, as shown in Fig. S4, which is consistent with the temperature dependent carrier concentration and the recent report”. However, in Fig. 1(d), we can clearly observe the nonlinear change of the carrier concentration vs. temperature, which plays the key role in the non-monotonical behavior.

Response: We apologize for the misleading statement in the previous version. Our ARPES measurements have confirmed that the Fermi level position and the Rashba constant (constant Rashba energy and momentum offset) keep almost unchanged with varying temperature. This is also in consistent with the fact that the absolute change of the temperature dependent carrier concentration remains weak in the magnitude, as shown in Fig. 1(d). However, as shown in Fig. S4, the increase of the temperature results in the spectral broadening, i.e. thermal broadening of the Fermi edge. It is accordingly inferred that the nonlinear variation of the carrier concentration with temperature originates from density of occupied states modulated by the temperature dependent Fermi-Dirac distribution function (thermal broadening).

The Fermi-Dirac distribution function is given by,

$$f_0(E) = \frac{1}{1 + e^{(\mu - E)/k_B T}}.$$

Fig. R8 shows the so-called thermal broadening of the Fermi edge with the constant Fermi level position. At $T=0$ K, the hole-type carrier occupation states are completely above the Fermi level μ . At $T>0$ K, $f_0(E = \mu) = \frac{1}{2}$ at the Fermi level position, and the occupation probability increases with increasing energy. Assuming that both the Fermi level position and the Rashba constant remain unchanged with the changing temperature, we theoretically calculate the temperature dependence of the carrier concentration via $n = \int \frac{f_0 d^3 \mathbf{k}}{(2\pi)^3}$ considering the Rashba-type band structures. As shown in Fig. 1d in the main text, the theoretical temperature dependent carrier concentration follows well the experimental result. As a consequence, it is verified experimentally and theoretically that the nonlinear variation of the carrier concentration with temperature originates from density of occupied states modulated by the temperature dependent Fermi-Dirac distribution function.

To make our description more transparent, we have revised Line106-109 (SI) to “It is seen that the Fermi level position and the Rashba constant in α -GeTe keep almost

unchanged with varying temperature, as shown in Fig. S4, which is consistent with the recent report⁴. This is also in consistent with the fact that the absolute change of the temperature dependent carrier concentration remains weak in the magnitude, as shown in Fig. 1(d).” in the revised manuscript.

Fig. R8 Scheme of the Fermi-Dirac distribution function at different temperatures for hole-type carriers.

3) Since “the Fermi level position and the Rashba constant in α -GeTe vanishingly vary with the temperature”, the authors neglected the temperature dependence of Rashba SOC for simplicity. Although Rashba SOC itself is vanishingly vary with the temperature, is it possible that some physical quantity is sensitive to Rashba SOC. As the authors pointed, the Fermi level is also vanishingly vary with the temperature, however, the sensitivity of J_x^2 to the Fermi level position plays the critical role in the non-monotonical dependence on temperature. If the author could reveal the nonreciprocal transport behavior from 0K to T_c of GeTe ($\sim 700K$), the investigation would be of more significance and instructive.

Response: We thank you for the suggestions/comments to improve our work. To make our description clearer, we elaborated the situation of the Rashba state in the applied magnetic field. The Rashba state without the applied magnetic field is shown in Fig. R9a. The band-crossing point is defined as the zero-point energy. The applied magnetic field gives rise to an energy shift with a value of $g\mu_B|B_y|$, as shown in Fig. R9b. Here, we emphasize that the nonreciprocal transport depends on the “relative position” of the Fermi level μ reference to the valence band maximum ($-\frac{m\alpha_R^2}{2\hbar^2}$) in the Rashba-type band, and can be simplistically described by

$$J_x^2 \sim \frac{1}{\sqrt{-\frac{m\alpha_R^2}{2\hbar^2} - \mu}} * F[f_2(T, \mu, \alpha_R)].$$

Here, $F[f_2(T, \mu, \alpha_R)]$ is the function of the second order correction $f_2(T, \mu, \alpha_R)$ to f_0 . We agree with that the both Fermi level position and Rashba SOC contribute to the nonreciprocal transport in GeTe. Our ARPES measurements have shown both the Fermi level position μ and the Rashba constant α_R almost keep unchanged, but the increase of temperature results in the thermal broadening of the Fermi edge. The non-monotonical variation of nonreciprocal transport with temperature (γ vs. T) originates from the density of occupied states modulated by the temperature dependent second-order Fermi-Dirac distribution function in α -GeTe. Based on the fact that μ and α_R are closely related, it is difficult to separately distinguish the contribution to the nonreciprocal transport from Rashba SOC.

In our work, we further calculated the γ vs. T at different Fermi level μ at a fixed Rashba SOC. It is found that the γ vs. T displays different characteristics if the Fermi level μ locates at different position in the Rashba-type state, as shown in Fig. 4d.

You also suggested to explore the nonreciprocal transport across the transition temperature. It is very interesting and significant to explore the Rashba SOC dependent nonreciprocal response. The Rashba SOC of the rhombohedral α -GeTe originates from the structural distortion between Ge and Te sublattices. GeTe undergoes a structure transition from the rhombohedral structure (space group R3m, noncentrosymmetry) to the cubic structure (space group Fm $\bar{3}$ m, centrosymmetry) at ~ 700 K. The Rashba SOC should also undergo a transition across the transition temperature. According to the previous works [*Phys. Rev. B* 82, 155209 (2010), *Appl. Phys. Lett.* 99, 231907 (2011), *Phys. Rev. B* 97, 224106 (2018)], the rhombohedral angle and the lattice parameter of GeTe increase upon increasing temperature and the dramatic variations of the parameters occur at high temperature (600-700K).

Following your suggestion, we further performed the measurement of the nonreciprocal charge transport above 300 K using the PPMS system (maximum magnetic field 9 T, maximum temperature 400 K). As shown in Fig. R10a, the nonreciprocal transport behavior in α -GeTe still exists at 375 K. The γ vs. T still displays the same trend as shown previously in Fig. R 10b. We are sorry that we do not have a system with both large magnetic field and high temperature to measure the nonreciprocal charge transport, and that we could not find such a system that we can access. In addition, according to the above formula, the nonreciprocal transport depends on the “relative position” of the Fermi level μ reference to the valence band maximum ($-\frac{m\alpha_R^2}{2\hbar^2}$) in the Rashba-type band. It is very difficult for us to perform the

high-temperature ARPES measurements, and further determine the Fermi level position and the Rashba constant. We will keep such opportunities in mind and pursue such measurements through collaboration in the future.

We have added the measured results above 300 K in Fig. S14 in the revised Supplementary Information.

Fig. R9 Rashba states without a and with b the magnetic field.

Fig. R10 a In-plane angle-dependent second-order harmonic longitudinal resistance ($R_{2\omega}$) at varied magnetic fields. The inset is the extracted magnetic field dependent $\Delta R_{2\omega}$. **b** Temperature dependent nonreciprocal coefficient γ with the theoretical prediction.

4) The author should check all the formula in the manuscript and SI, there are many missing symbols, for example, line 44, 68, 89.....

Response: Thank you very much for pointing out the compatible issues between different versions of Microsoft Office. We have checked and corrected the missing symbols in all the formula thoroughly.

5) On line 415, there is a space before the word “induces”.

Response: Thank you for the very careful reading. We have corrected this typo in the

revised manuscript.

REVIEWERS' COMMENTS

Reviewer #1 (Remarks to the Author):

The authors responded my comments appropriately. I now recommend its publication in the present form.

Reviewer #2 (Remarks to the Author):

I feel that authors answered all the questions appropriately and improved the manuscript by adding new discussion. I think nonreciprocal charge transport at room temperature reported in this manuscript is scientifically interesting and important for the application, attracting broad attention.

Reviewer #3 (Remarks to the Author):

All the comments have been well responded. The revised version of the manuscript is now suitable for publication in Nature Communications.